



# Quantifying unequal urban resilience to rains across China from location-aware big data

Jiale Qian[1,2], Yunyan Du[1,2,*], Jiawei Yi[1,2], Fuyuan Liang[1], Nan Wang[1,2], Ting Ma[1,2], Tao Pei[1,2]

[1]State Key Laboratory of Resources and Environmental Information System, Institute of Geographical Sciences and Natural
Resources Research, Chinese Academy of Sciences, Beijing 100101, China.
[2]University of Chinese Academy of Sciences, Beijing 100049, China.

*Correspondence to*: Yunyan Du (duyy@lreis.ac.cn)

**Abstract.** The disaster-relevant authorities could make an uninformed decision due to the lack of a clear picture of the urban resilience to adverse natural events. Previous studies seldom examine the near-real-time human dynamics, which are critical

to disaster emergency response and mitigation, in response to the development and evolution of a natural hazard. In this study, we used the aggregated Tencent location request data (TLR) to examine the variations in collective human activities in response to rains in 346 cities in China. Then we report a comprehensive study of the urban resilience to rains across the mainland China. Our results show that, on average, a 1-mm increase in rainfall intensity is associated with a 0.49% increase of the human activity anomalies. In the cities of northwestern and southeastern China, human activity anomalies are affected more by rainfall

intensity and rainfall duration, respectively. Our results highlight the unequal urban resilience to rains across China, showing current heavy rain warning standards underestimate the impacts of heavy rains on the residents in the northwest arid region and the central underdeveloped areas, and overestimate the impacts on the residents in the southeastern coastal area. An overhaul of current heavy rain alert standards therefore is needed to better serve the residents in our study area.

## 1 Introduction

Heavy rains with intense precipitation have become more frequent in the context of global climate changes(Myhre et al., 2019; Ogie et al., 2018) and pose significant threats to urban residents, mainly due to the uncoordinated watershed management and undersized infrastructures(Chan et al., 2018; Dewan, 2015; Nahiduzzaman et al., 2015; Song et al., 2019). China is frequently perplexed by urban flooding, particularly in summer when the Asian monsoon brings heavy rains to inland China. It is estimated that 55.15 million people are affected by floods in China in 2017 alone and the direct economic loss is approximately

214 billion Chinese Yuan, which significantly exceeds that of the 2017 typhoon disasters (5.879 million, 34.62 billion). In addition to threatening human daily activities and cities' normal operation(Aerts et al., 2014; Grinberger and Felsenstein, 2016; Kasmalkar et al., 2020; Owrangi et al., 2014), the ever-increasing rainstorms endlessly challenge cities' flood resistance capacity and relevant authorities' real-time decisions in response to such adverse events. Urban decision makers have learned that city management and planning would significantly benefit from a better understanding of urban resilience(Bertilsson et

al., 2019; de Bruijn, 2004; O'Sullivan et al., 2012).





Urban resilience refers to the ability of an urban system to prepare for, respond to, and recover from adverse events(Ambelu et al., 2017; Hong et al., 2021; Liao, 2012; Meerow et al., 2016). Biologists, psychologists, engineers, and geographers have all made their own contributions to the urban resilience studies(Adger et al., 2005; Brusberg and Shively, 2015; Olsson et al., 2015; Ouyang et al., 2012; Poulin and Kane, 2021; Shiferaw et al., 2014). Over the past 20 years, geographers have heavily 35 relied satellite imagery to assess disaster-related resilience as satellites have been providing ever-increasing information about the Earth at a relatively low cost(Mpandeli et al., 2019; Stefan et al., 2016; Tellman et al., 2021). For example, satellite-based emergency mapping systems have been developed to monitor the inundation and recovery processes of the 2005 Switzerland flood(Buehler et al., 2006); assess the damage, restoration, and reconstruction induced by the 2010 Haiti earthquake(Honey et al., 2010); and evaluate the changes of power supply before and after Hurricane Maria in 2017(Román et al., 2019). Emergency 40 rescuers can use high-resolution images to closely monitor ongoing natural disasters and coordinate disaster relief. However, it is almost impossible to extract near real-time human dynamics over the evolution of a disaster from satellite images and such information is very important in disaster mitigation and reduction(Ghaffarian et al., 2018; Liu et al., 2015).

Location-aware big data such as the smartphone call records, signaling data, and social media posts, have been widely used to infer real-time human activities(Jiawei Yi et al., 2019; Wang et al., 2020, 2019; Yue et al., 2017), estimate disaster-induced 45 losses(Kryvasheyeu et al., 2016; Zhang Liu et al., 2019), monitor resettlement and restoration(Martín et al., 2020a, 2020b; Wang and Taylor, 2018; Yabe et al., 2020), and study disaster-related resilience(Hong et al., 2021; Huang and Ling, 2018; Kasmalkar et al., 2020; Zou et al., 2018). Urban residents would adjust their activities when their living environments are socially and physically impacted by an adverse event and such adjustment could be inferred from location-aware big data. In other words, the changes of human social activities extracted from location-aware big data (hereafter referred to as social 50 activities unless specified otherwise) could be used to study the resilience capacity of an urban system in response to an adverse event.

Human social activities may change in response to a disaster and quite a few studies have been conducted to evaluate the urban resilience to destructive disasters(Qiang and Xu, 2020; Takagi et al., 2021; Walch, 2018). For example, significant social and geographic differences were found in the use of social medias across the 126 counties impacted by Hurricane Sandy(Zou et 55 al., 2018) and the 76 counties in Texas and Louisiana affected by Hurricane Harvey(Zou et al., 2019), illustrating how residents prepare for, respond to, and recover from such catastrophic events.

Human social activities may also change in response to mild yet frequent adverse natural events, such as urban floods. Unlike Hurricanes, dwellers are usually not mobilized by relevant authorities to prepare for and resettle after such events. Instead, nearly 90% of flood-related tweets in a city are released during heavy rains(Wang et al., 2020). Consequently, social media 60 activities mainly show how an urban system respond to but not prepare for and recover from such adverse natural events. Furthermore, the mild adverse natural events usually affect a large geographic scale and for a much longer time. As a result, urban resilience to mild and frequent adverse events could significantly differ from that of destructive disasters and may show significant spatiotemporal variations due to the areal difference in local natural settings, socioeconomic status, and





infrastructure completeness(Adger et al., 2005; Guan and Chen, 2014; Östh et al., 2015; Zou et al., 2019, 2018). Study of such regional inequality is therefore of great value to disaster relief and mitigation.

In this study, we proposed a method for measuring and evaluating how urban systems respond to heavy rains as reflected in location-aware big data. We extracted human social activities from the Tencent location request (TLR) data in 346 cities across mainland China from May to August 2017 and used two indicators - rainfall threshold and response sensitivity – to quantify the urban resilience across our study area. We found significant regional inequality of urban resilience in mainland China and the inequality could be explained by the variations in the regional natural, socioeconomic, and infrastructure variables. The findings from this study provide a new perspective and method to quantify urban resilience to the frequent yet no so destructive adverse events across a large geographic scale. Practically, our findings suggest an urgent need to revise current unified rainstorm warning standards to better serve the residents.

## 2 Data and methods

### 2.1 Data

We collected the Tencent location requested (TLR) data from May $1^{st}$ to August $31^{st}$, 2017 from Tencent's big data portal. Tencent, with over 700 million users, is the largest social media platform in China. A Tencent user may check in the platform for a variety of purposes such as location-based searching, navigation, location sharing and so on. The dataset we downloaded has an hourly temporal resolution and a 1 km x 1 km spatial resolution. The data has been proven as a reliable proxy for collective human activities in many studies from multiple dimensions of time and space(Ma, 2018; Qian et al., 2021; Zhang Liu et al., 2019).

We collected the Version06 Global Precipitation Measurement (GPM) Integrated Multi-satellite Retrievals (IMERG) 30-min precipitation dataset(Levizzani et al., 2020, p. 1). This dataset has a spatial resolution of $0.1° \times 0.1°$ and has been evaluated and widely used in related studies(Jiawei Yi et al., 2019; Liu et al., 2019). We used this dataset to extract the characteristics of rainfall events of interest.

We collected six natural, socioeconomic, or infrastructure indicators to help explain the variations in urban resilience, including the annual precipitation in China since 1980, population density, gross domestic product, green coverage rate, drainage network density, and per capita area of paved roads. The precipitation data was obtained from https://www.resdc.cn/data.aspx?DATAID=229 and the other indicators from the 2017 China City Statistical Yearbook.

### 2.2 Methods

Fig. 1 shows the data process and analysis flow chart of this study. We first proposed a Multilevel Human Activity Anomaly Detection (MHAAD) methodological framework to detect and characterize the TLR anomalies in response to rainfall events. The framework has two major parts. In the first part, we identified the grids with a stable TLR number and then the anomalies from the time series TLR of each grid. We then used the two-sided Welch's t-test and probability density function (PDF)




method to detect whether human activity anomalies are triggered by a rainfall event or not. Rainfall indices were extracted for

the grids with a stable TLR number and selected by their importance as shown in the random forest model. We then explored

the multi-level relationship between rainfall characteristics and human activity anomalies. We proposed two indicators -rainfall

threshold and response sensitivity- to describe urban resilience. Lastly, we assessed the association between the urban

resilience with the natural, socio-economic, and infrastructure explanatory variables.

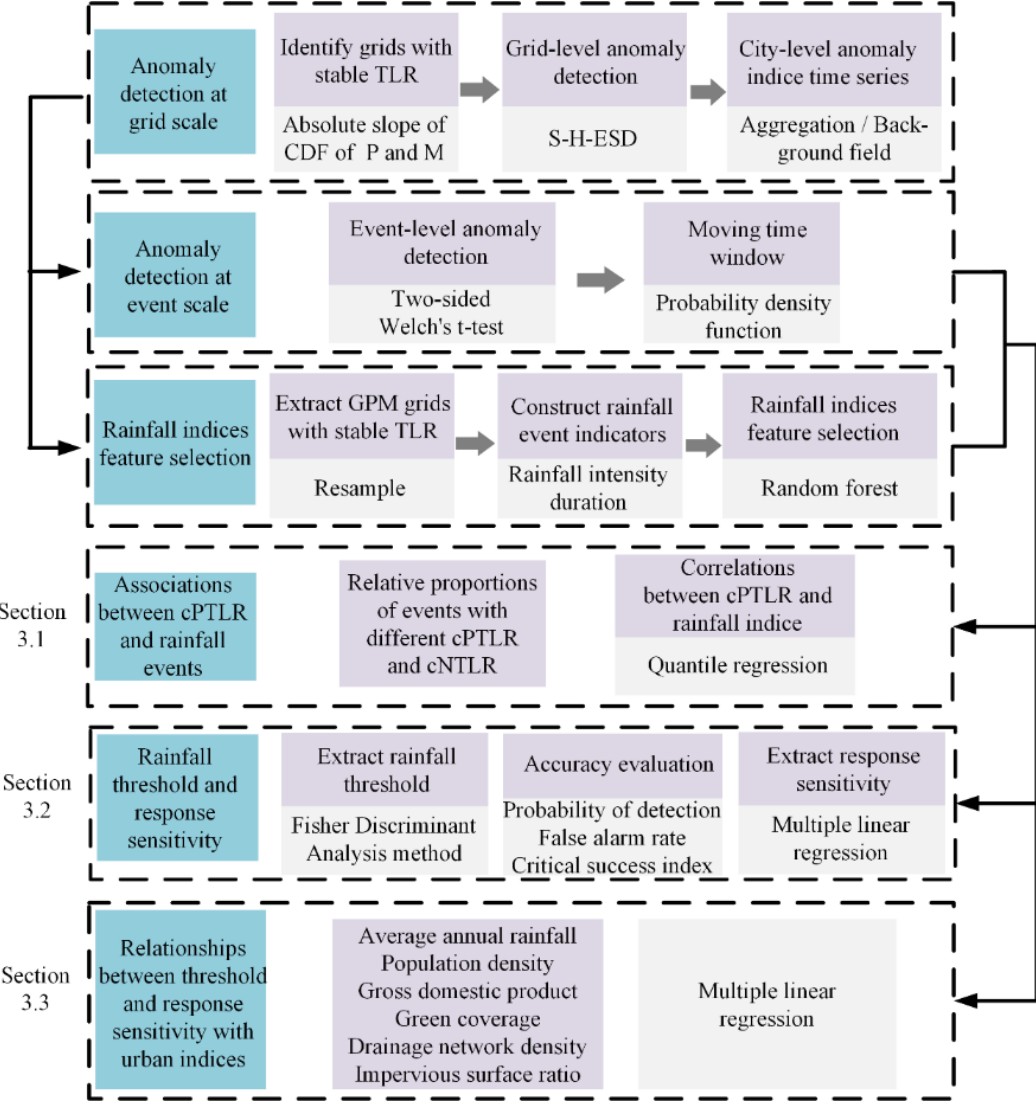


**Fig. 1 A flow chart showing the data analysis process in this study.**

## 2.2.1 TLR anomaly detection

We first identified all grids with a stable TLR number (hereafter are referred as to stable grids unless stated otherwise) using

the method Qian et al.(Jiale et al., 2021) proposed. In total, 832,630 out of the 9,600,903 grids across our study area are




identified as stable grids. The number of smartphone users and TLR vary significantly by grid. We therefore normalized the TLR using the median interquartile normalization method(Geller et al., 2003) to make the TLR in the stable grids comparable. We then employed the Seasonal Hybrid Extreme Studentized Deviate (S-H-ESD) method(Vallis et al., 2014) to detect anomalies from the gridded TLR time series. The S-H-ESD method could be denoted by the following additive model:

$$T_s = T + S + R \tag{1}$$

where $T$, $S$, and $R$ represent the trend, seasonality, and residual components, respectively.

The S-H-ESD method has two major steps. First, the piecewise median method is used to fit and remove the long-term trend. Then the Seasonal-Trend decomposition using Loess (STL) method is employed to remove seasonality(Cleveland et al., 1990). We then used the Generalized Extreme Studentized Deviate (G-ESD) statistic(Rosner, 1975) to identify the significant anomalies in the residuals. In this study, we used a piecewise combination of the biweekly medians to model the underlying

trend, which shows little changes in the TLR time series. The significance level $a$ is set to 0.05 and the number of anomalies is set to no more than 25% of the total observations.

### 2.2.2 Feature selection of rainfall indices

The GPM data were first resampled to the same spatiotemporal resolutions as that of the TLR using the nearest-neighbor interpolation method. We then extracted the hourly rainfall intensity, i.e., the average GPM precipitation within the stable

grids.

In this study, a rainfall event is defined as a precipitation process that lasts for at least 3 hours and with no rains preceding for at least one hour. The number of rainfall events in each city is normally distributed. We selected the 346 cities with at least 40 (the top 5% quantile) rainfall events for this study (Supplementary Fig. 1 and 2).

Every rainfall event is described with three rainfall indices (the 1-hour peak intensity, 6-hour peak intensity, and cumulative

rainfall) and two temporal indices (the duration and peak hour) (Supplementary Table 1). From these indices, we used the random forest model (RF) to calculate the importance score (mean decrease accuracy) for each indicator. The importance score shows the global importance over all the out-of-bag cross validated predictions. The random forest model is robust and less susceptible to multicollinearity as it averages all predictions for a given feature variable and is more efficient in terms of feature selection than the multi-linear regression(Pal, 2005; Strobl et al., 2007). We then identified the most important indicator that

triggers human activity anomalies(Liaw and Wiener, 2001).

### 2.2.3 Anomaly detection at event scale

We first extracted the total numbers of the grids with positive (PTLR) and negative anomalies (NTLR) by city, respectively and then examined the variations in the PTLR/NTLR time series over the periods with rains and without rains to identify whether a rainfall event triggers collective human activity anomalies. The two-sided Welch's t-test was used for the

significance test. Human activity anomalies usually happen shortly before or after the peak rainfall intensity and last as a spell instead of the entire rainfall event. We therefore employed a moving time window method to find the period with the largest



accumulative rainfall and used the period to detect the statistical significance of the change in related to the PTLR/NTLR time series in a raining and non-raining period. In this study, we used a 6-hour moving window, which is half of the average duration of all rainfall events of interest in this study (see Supplementary Fig. 3).

**2.2.4 Quantifying the rainfall threshold and response sensitivity**

In this study, we used two indices, rainfall threshold and response sensitivity, to quantitatively characterize urban resilience. We first used a linear binary classifier to examine the paired values of the peak intensity and duration to determine whether a rainfall event brings more or less rain than the threshold to trigger collective human activity anomalies (Fig. 2a). The Fisher Discriminant Analysis method was then applied to identify the discriminant function to minimize the classification errors(Mika 145 et al., 1999). The rainfall thresholds associated with different rainfall durations are directly extracted from Fig. 2a.

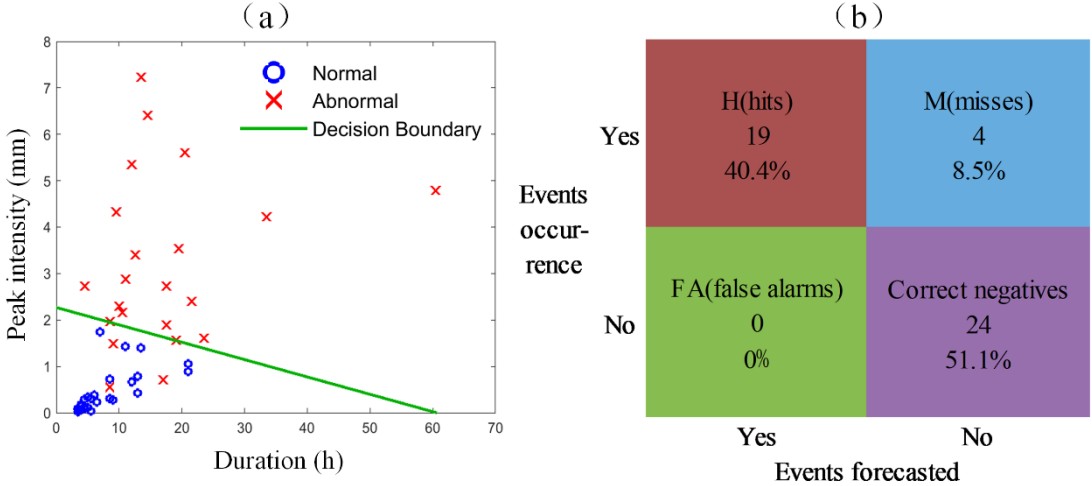

Fig. 2 The schematic diagram (a) and contingency table (b) of the binary classifier for Beijing.

Three quantitative indices, the probability of detection (*POD*), false alarm rate (*FAR*), and critical success index (*CSI*) are used 150 to evaluate the performance of the threshold identification method based on the contingency table (Fig. 2b):

$$POD = \frac{H}{H + M} \tag{2}$$

$$FAR = \frac{FA}{H + FA} \tag{3}$$

$$CSI = \frac{H}{H + M + FA} = \frac{1}{POD^{-1} + (1 - FAR)^{-1} - 1} \tag{4}$$

where H, FA, M represent the percent of hits, false alarms, and misses, respectively. All three indices range from 0 to 1. A 155 value of 1 for the POD and FAR indicates a perfect hit and a 100% false positive rate. A higher CSI is associated with a higher POD and a lower FAR value.





The second index, the response sensitivity, is defined as the rate of abnormal change in collective human activities triggered by a rainfall event. We first selected the rainfall events with precipitation above the threshold and that trigger human activity anomalies, i.e., those represented with red crosses and are above the decision boundary in Fig. 2a. Then we defined two

indicators cPTLR/cNTLR to describe the rate of abnormal change in collective human activities (Supplementary Fig. 4). The cPTLR/cNTLR were calculated as the difference between the mean of the two time series, i.e., the PTLR/NTLR in the 6-hour raining time window and non-raining time window. A multiple linear regression model was then constructed between the cPTLR and the rainfall intensity/duration for each city. In the end, we calculated the marginal city-specific partial derivatives of cPTLR with respect to the peak intensity and the duration, respectively. The response sensitivity index is calculated as the

average of the regression coefficients of the peak intensity and the duration. The adjusted $R^2$ was also calculated to assess the model accuracy.

Finally, we separately classified the rainfall threshold and response sensitivity indices of the 346 cities into three classes using the Jenks natural breaks classification method, which clusters data into different groups by seeking minimum variance within a class and maximum variance between classes(McDougall and Temple-Watts, 2012). In this study, we used the 6-hour rainfall

threshold index in line with the time window in Supplementary Fig. 4. In total, there are nine different combinations between the threshold and response sensitivity indices.

### 2.2.5 Quantifying the relationships between resilience and urban characteristics

We then examined the relationships between the two urban resilience indices and the annual rainfall, population density, gross domestic product, green coverage rate, drainage network density, and per capita area of paved roads. The Kendall, Pearson,

and Spearman correlation coefficients and multi-linear regression were used to measure the correlation between rainfall threshold, response sensitivity, and the city characteristics at the city level, respectively.

## 3. Results

### 3.1 Collective human activities in response to rainfall events

The gridded TLR could increase (positive anomaly) or decrease (negative anomaly) in response to rains (Supplementary Fig.

5). Counting the overall TLR changes by city would not show how rains impact collective human activities(Jiawei Yi et al., 2019). In this study, instead, we calculated the changes of the total numbers of the grids showing positive (cPTLR) and negative anomalies (cNTLR) by city during a raining period in related to those over the non-raining period, respectively (Fig. 3a) to illustrate how collective human activities change in response to rains.

The city-level collective human activities jump to an excited state (Fig. 3b) with a significantly increased number of the grids

exhibiting positive anomalies, in response to 55.11% of the daytime rains (Supplementary Fig. 6) whereas nighttime rains show no significant impact on the collective human activities. About 93.2% (i.e., 10,710) of the rainfall events in this study are associated with a greater change of the number of the grids with positive anomalies than that of the grids with negative





anomalies (i.e., |cPTLR|>|cNTLR|). Around 59.7% of the 10,710 (i.e., 6394) rains show an increased number of the grids with positive anomaly by city. Furthermore, 35.3% of the 6,394 (i.e., 2257) rainfall events associated with excited-state human activities show a significant increased number of the grids with positive anomalies, which we believed could be attributed to heavy rains. We noticed that a small number (103, 13.19% |cPTLR|<=|cNTLR|) of heavy rains brought by typhoons trigger the city-level collective human activities to a dispirited state, with a significantly increased number of grids exhibiting NTLR (Fig. 3c) as compared to the non-typhoon rains. Accordingly, we excluded all typhoon-related rainfall events from this study. The higher rainfall intensity could trigger more excited-state collective human activities (Fig. 3d). The 1-hour peak intensity values of the rainfall events associated with excited-state human activities are positively correlated with the corresponding cPTLR values (fitting slope = 0.49%, p value <0.001). However, the cPTLR slope against rainfall is affected by the divergence of the peak intensity anomaly. Quantile regression results show that the cPTLR slope coefficient estimates gradually increase from 0.17% for the lower 25% quantile to 0.84% for the higher 75% quantile (p value <0.01). In other words, the cPTLR growth rate generally increases from the less-anomaly to the sensitive-anomaly rainfall events with respect to the increasing magnitude of the rainfall peak intensity.

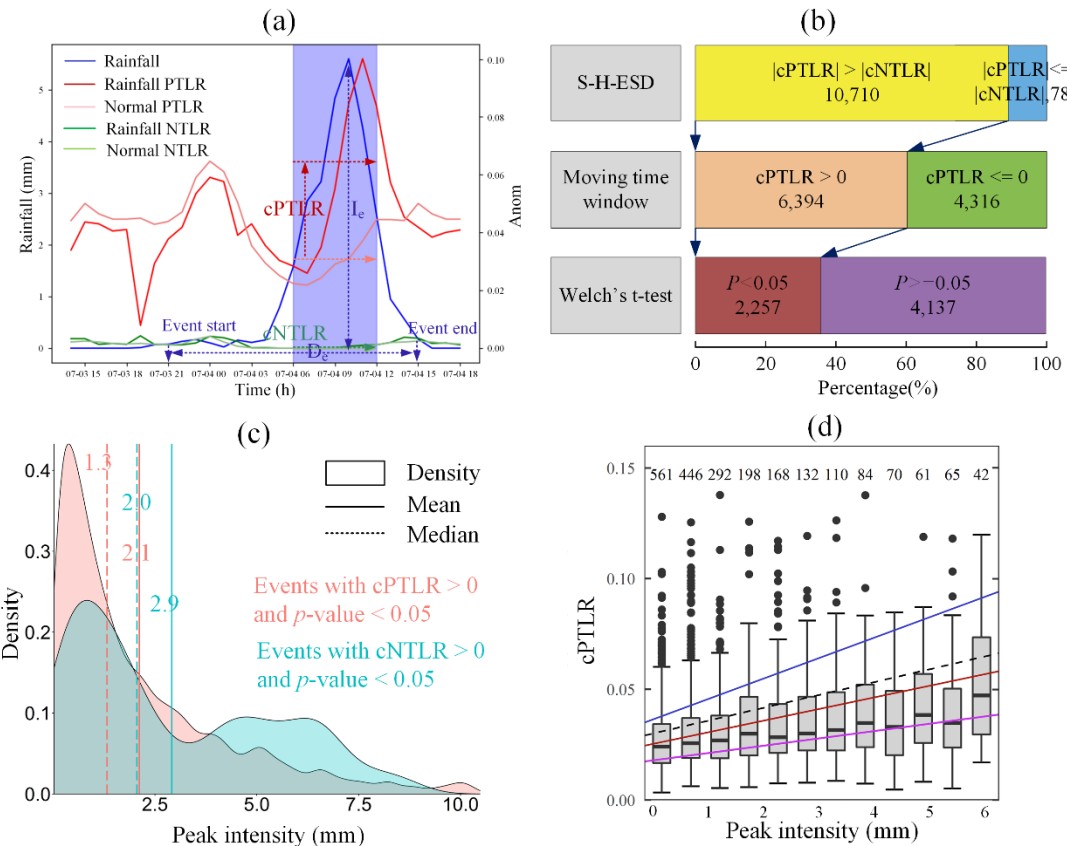

**Fig. 3 (a) The schematic definitions of the human activity anomaly associated with a specific rainfall event used in this study. (b) The relative proportions and numbers of the rainfall events with various cPTLR and cNTLR values. (c) The probability density**





**distribution curves of the events with significantly increased cPTLR/cNTLR. (d) Box plots by peak rainfall intensity groups per**
**0.5mm. Trend lines are shown for the OLS regression (black dotted), 0.25 (purple), median (red), 0.75 quantile (blue) range. The**
**numbers in Fig. 3(d) show the numbers of rainfall events with specific peak intensity as shown along the x axis.**

### 3.2 Regional inequality of urban resilience

We derived two indicators - rainfall threshold and response sensitivity - from the perspective of public's social response to
rains to evaluate the urban resilience. The threshold is defined as the intensity of an event that triggers a city into an undesirable
state(Liao, 2012). In this study, we defined the threshold of a rainfall event with a specific duration as the minimum rainfall
intensity that corresponds to significant urban TLR anomalies. As shown in Supplementary Fig. 6a, the peak rain intensity and
rain duration are the second and third important characteristics that trigger human activity anomalies, respectively. We
extracted the rainfall thresholds for each city using binary classification models (supplementary Fig. 7) and the results show
that the rainfall thresholds drop with increased rainfall duration across all 346 cities in China (Fig. 4 a-c). The 3-, 6-, and 12-
hour rainfall thresholds all show significant spatial autocorrelation and a pattern of gradual decrease from the southeast coast
to the northwest inland. However, with increased rainfall durations, the average rainfall thresholds of all 346 cities decrease
from 4.24 to 2.75, whereas their standard deviations decrease from 3.55 to 2.45. Such results indicate the public's response to
the short rainfall events varies greatly in different cities and tends to be more consistent with increased rainfall durations.

The impacts of the peak rain intensity and duration on human activity vary across our study area as shown by the slopes of the
decision boundary of different cities (Fig. 4d). In the arid and semi-arid northwestern China, the slope is close to zero, showing
the public response is mainly affected by peak intensity. Residents in the northwestern China may adjust their activities in
response to the rain peak intensity as rains in this area seldom last long. By contrast, the slope is high for the southeastern
region, indicating the public's response is more affected by rainfall duration. The wet southeastern China usually receives
frequent and heavy rains and residents already have adopted to it. Consequently, residents in this area may change their
activities in response to rainfall duration more than the peak intensity.

Results of the binary classification are solid as shown by the anomaly detection POD, FAR, and CSI values for different
rainfall durations based on rainfall thresholds (Supplementary Fig. 8). More specifically, the POD values for different rainfall
durations in the 346 cities range from 0.71 to 1.00, the FAR values from 0 to 0.46, and the CSI values from 0.48 to 1.


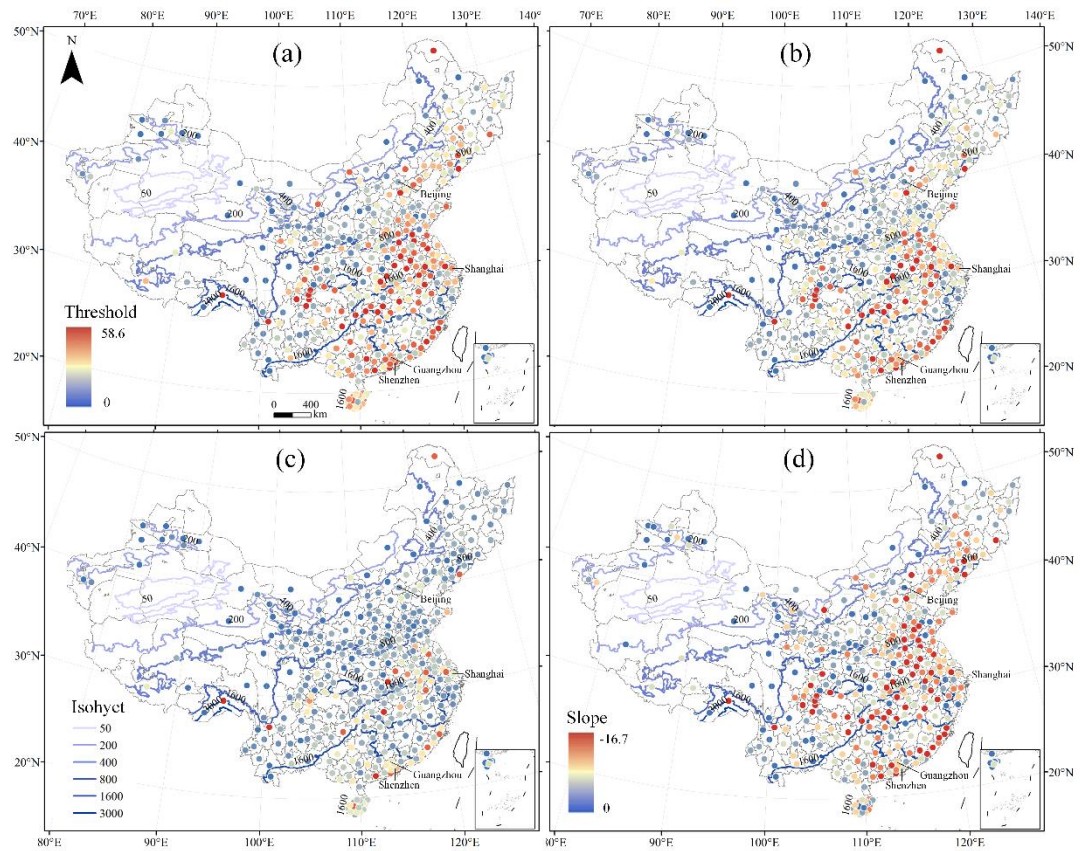

**Fig. 4 Spatial distribution of the rainfall threshold that triggers human activity anomalies for rainfall events lasting three (a), six (b), and 12 (c) hours, respectively and slope (d) of the decision boundary. The isohyets show the 2017 annual rainfall.**

The other urban resilience indicator, the response sensitivity, is defined as the rate of the collective human activity anomalies

triggered by a rainfall event and was extracted from the multiple regression analysis. The response sensitivity is low in the southeast coast and high in the northwest inland, showing an opposite trend as that of the rainfall threshold (Fig. 5a). The higher response sensitivity in the northwest suggests that the residents in this area tend to change their activities more significantly in response to rains. By contrast, activities of the residents in the southeast are not significantly impacted by rains. Such findings are consistent with those derived from the rain threshold indicator. The accuracy of the regression model (the

adjusted $R^2$) also shows a similar trend as that of the response sensitivity (Fig. 5b), which indicate that the response mode of collective human activities to rainfall in the southeast coastal area is more complex.


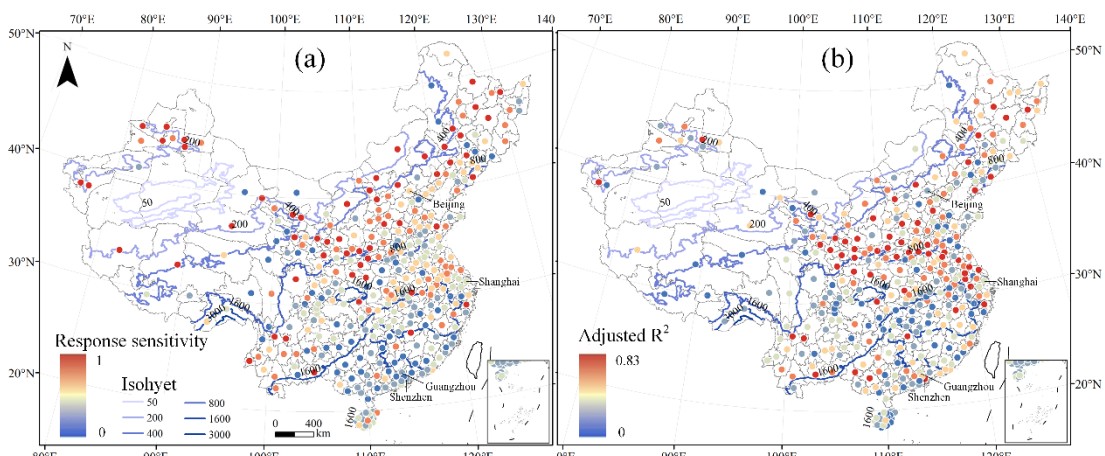

**Fig. 5 Spatial distribution of the response sensitivity (a) and adjusted R$^2$ (b) in multiple linear regression. The isohyets show the 2017 annual rainfall.**


The associations between the two urban resilience indices also exhibit a significant pattern across our study area (Fig. 6). Cities located in the area with over 1600mm annual precipitation are mainly categorized into type HL (threshold > 10.29mm and response sensitivity < 0.003), and surrounded by the types ML and HM cities. The cities located in the areas with less than 400mm annual precipitation are mainly classified into LH, MH, and LL types, indicating the annual precipitation has a

significant impact on the human activities in different cities.

The LH type cities have a low rainfall threshold (<3.25mm) and high response sensitivity (>0.025). These cities are mainly found in the northwest fragile region (mainly including the Yili prefecture in Xinjiang province) and the central underdeveloped China (including Shaanxi, Shanxi, and Hebei provinces). Such cities may have underdeveloped infrastructure and weaker rains could trigger human activity anomalies.


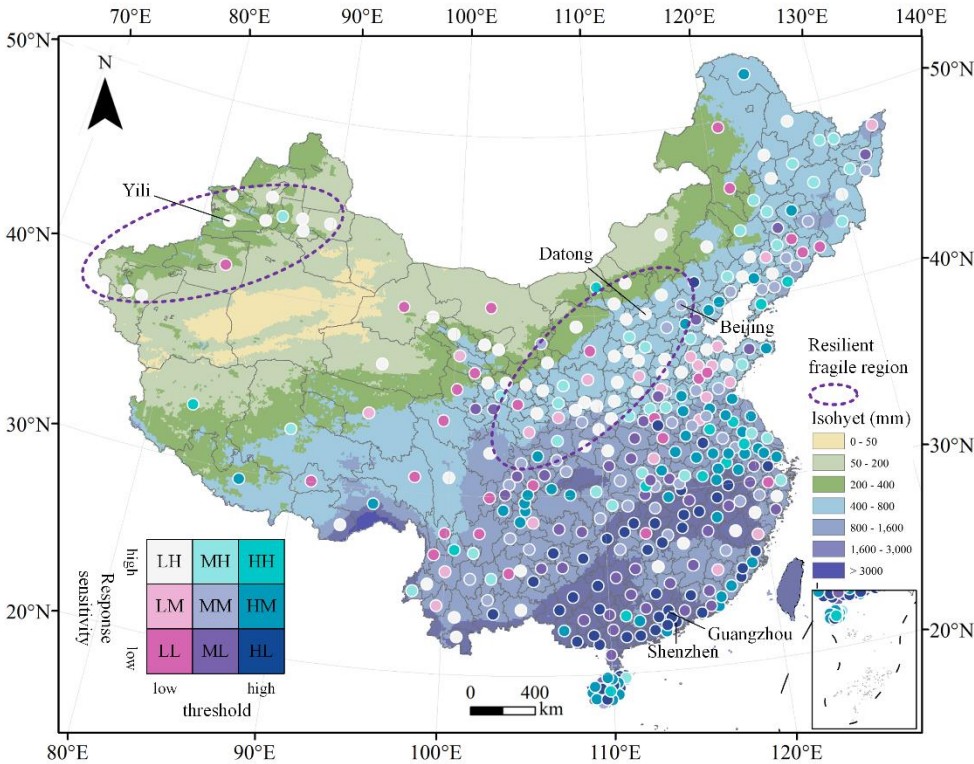

**Fig. 6 Spatial distribution of urban resilience. The basemap colors indicate the 2017 annual rainfall.**

### 3.3 Associations between the urban resilience with the urban indicators

Results of multiple regression analysis show that variations in urban resilience by city could be explained by the variations in a variety of natural, socioeconomic, and urban infrastructure indicators. Fig. 7a shows the relationship between rainfall thresholds and the explanatory indicators. The 3-hour, 6-hour, and 12-hour rainfall thresholds all show similar correlations with the indictors (Supplementary Table 2, 3, 4), and we only show the correlations between the 6-hour rainfall and the indicators in this section.

All six explanatory variables are significantly correlated with the rainfall threshold ($p < 0.05$). About 42% of the variations in the rainfall threshold could be explained by the variations in the explanatory variables as shown by the $R^2$ (Supplementary Table 3). Among all explanatory variables, the annual rainfall has the highest coefficient value of 0.43, indicating the variations in the threshold are most affected by the annual rainfall. In other words, residents living in regions with different annual precipitation amount are more likely to accordingly adjust their daily activities once the rainfall is over a specific threshold.

Other explanatory variables are also positively correlated with the threshold variable as shown by the positive correlation coefficients ranging from 0.21 to 0.10, except the per capita area of paved road, which is negatively correlated with the threshold. In fact, the per capita area of paved roads is the only indicator showing a negative correlation. Increased per capita





area of paved road weakens rainwater infiltration capacity and increases surface runoffs, which is more likely to cause traffic congestion and trigger human activity anomalies even the rainfall amount is below a lower threshold.

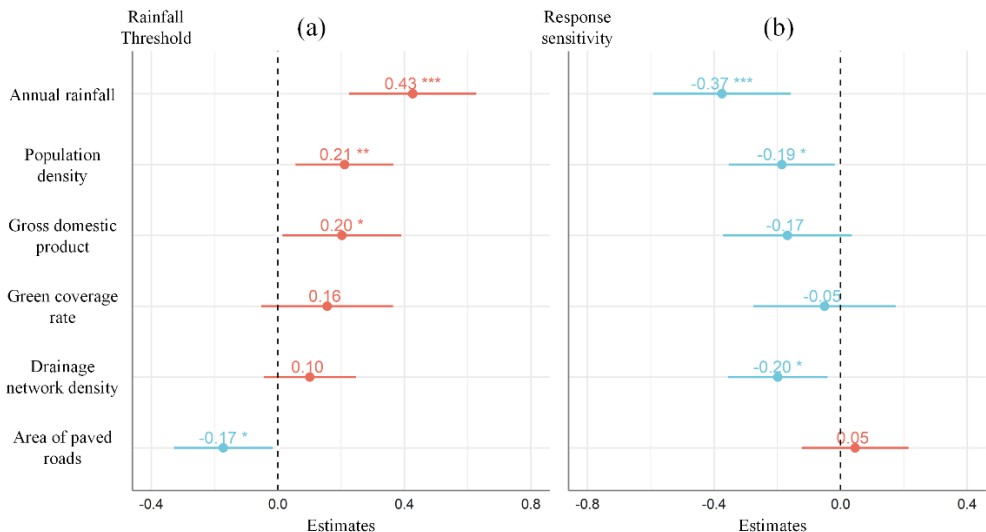

**Fig. 7 Regression coefficients between the six explanatory variables and the 6-hour rainfall threshold (a), response sensitivity (b). The horizontal lines mean 95% confidence interval.**

Multi-regression analysis shows that the response sensitivity is negatively correlated with all explanatory variables, except for the per capita area of paved roads, which show a positive correlation (Fig. 7b, Supplementary Table 5). All correlations are statistically significant ($p<0.05$). About 31% of the variations in the response sensitivity could be explained by the variations of the explanatory variables as shown by an $R^2$ value of 0.31.

## 4. Conclusions and discussions

Residents in different cities may adjust and change their activities in different ways in response to rains. Such activity changes and adjustment, also known as urban resilience, could be characterized and studied from location-aware big data. In this study, we used the Tencent aggregated location request data to examine the changes of collective human activities in major cities in China in response to rains over 2017 summer. Our results show that rainfall time, peak intensity, and duration are the three most important indicators that determine whether a rain would trigger human activity anomalies or not. We also proposed two indices, the rainfall threshold, and response sensitivity, to describe the urban resilience, which show significant spatial variations across our study area. Furthermore, the unequal urban resilience could be explained by a series of explanatory variables.



We believe this paper provides a new perspective to studying urban resilience and that the results bridge the knowledge gap between heavy rains, collective human activities, and urban resilience. Such knowledge is of great significance to urban planning, traffic management, and emergency response.

We also believed this study has three other contributions regarding urban resilience research. First, this study expanded the research framework of urban resilience in response to high-frequency yet mild adverse events and such a framework could be

used to study the variations in urban resilience across a large area. Previous studies mainly focused on urban resilience to a specific adverse event such as a typhoon or a hurricane. For example, Zou et al.(Zou et al., 2018) used a normalized ratio index to assess the regional variations in urban resilience to Hurricane Sandy. In this study, we examined urban resilience to rains over a relatively long period and across a large area. Such a study would better show how residents in different regions change their activities in different ways in response to rains with different duration, peak intensity, and accumulative rainfall.

Secondly, our research analyzed the impacts of different features of the rainfall events on human activities. Previous studies often simply characterized a disaster using its threat levels. For example, Zou et al.(Zou et al., 2018) used the average hurricane track kernel density and its wind speed to define the threat levels. In this study, instead, we extracted five major elements of a rainfall event and employed the random forest model to study the impacts of different elements on collective human activities. Thirdly, we used the rainfall threshold to quantify urban resilience and the thresholds is valuable for the authorities to revise

heavy rain alerts. Conventionally, heavy rain alerts are usually based on rainfall intensity and precipitation only and seldom consider the areal difference of infrastructures. According to current Chinese Standard, Chinese authorities would issue a blue, yellow, and orange alert when precipitation is or will be over 50 mm in 12, 6, and 3 hours and if the rain might not stop(Mendiondo, 2005). A red alert would be issued when precipitation is or will be over 100 mm in 3 hours and if the rain might not stop. Results from this study show that it is not appropriate to apply such a unified alert standard to different groups

of cities across China. In the resilience fragile areas (Fig. 6), for instance, a rainfall event with 3.25mm precipitation per hour (i.e., 19.5mm in six hours) already triggers significant human activity anomalies. As a result, the national heavy rain alert standard significantly underestimates the impacts of rainfall on the residents in the northwestern and central China.

Tab. 1 shows the precipitation thresholds of different city groups that trigger human activity anomalies and the 6-hour accumulative precipitation based on which a heavy rain warning should be issued. For example, for the LH cities, a rainfall

intensity of 1.8 mm and 10.8mm 6-hour accumulative precipitation should trigger a yellow heavy rain warning. Such a value of the 6-hour accumulative precipitation is much lower than current Chinese heavy rain yellow warning standard (50 mm in 6 hour). By contrast, for the HH cities, the rainfall threshold and 6-hour accumulative precipitation that trigger human activity anomalies is 22.34 mm and 134.05 mm, respectively. The accumulative precipitation is much higher than the heavy rain yellow warning standard. In other words, it is amateurish to issue a heavy rain warning when the 6-hour accumulative precipitation is

50 mm for the HH cities. Results from this study therefore are of great value for the authorities who revise heavy rain alerts across China to help local residents be better prepared for such adverse events.



**Tab. 1 Urban resilience indecies from this study for different city groups and the difference from current Chinese yellow heavy rain alter standard.**

| Type | Threshold(mm) | Response sensitivity | Alarm standard(mm) | Yellow warning(mm) | Difference from yellow warming |
|------|---------------|----------------------|--------------------|--------------------|-------------------------------|
| LH | 1.8 | 0.71 | 10.8 | 50 | -39.2 |
| LM | 2.08 | 0.51 | 12.46 | 50 | -37.54 |
| LL | 2.23 | 0.39 | 13.35 | 50 | -36.65 |
| MH | 3.61 | 0.63 | 21.68 | 50 | -28.32 |
| ML | 4.99 | 0.41 | 29.94 | 50 | -20.06 |
| MM | 5.03 | 0.49 | 30.19 | 50 | -19.81 |
| HL | 17.5 | 0.42 | 104.97 | 50 | 54.97 |
| HM | 18.94 | 0.48 | 113.64 | 50 | 63.64 |
| HH | 22.34 | 0.64 | 134.05 | 50 | 84.05 |


**Data availability**

The information can be made available upon request to the corresponding author.

**Author contributions**

J.Q., Y.D., J.Y., and F.L. conceived and designed the study and methods; J.Q., J.Y., Y.D., N.W., T.M., and T.P. analyzed the
data; J.Q., F.L., Y.D. and J.Y. wrote the paper, and all coauthors contributed to the interpretation of the results and to the text.
All authors read the manuscript and approved the submission.

**Competing interests**

The contact author has declared that neither they nor their co-author has any competing interests.

**Financial support**

This research was jointly supported by the Strategic Priority Research Program of the Chinese Academy of Sciences (Grant
No. XDA19040501), the National Key Research and Development Program of China (Grant No. 2017YFC1503003), the
National Science Foundation of China (42176205 and 41901395).



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
