# Peer review of "Quantifying unequal urban resilience to rainfall across China from location-aware big data"

_Natural Hazards and Earth System Sciences, 2022_

## Author Response (AR1)

Specific responses are as follows:

1.  The flowchart in Fig.1 is too complicated to follow the key information. It should be simplified to present the key information.

Response: Thanks for the comment. We simplified the flowchart in Fig.1 and adjust the corresponding text to present the key information. Specific information is as follows:

[Figure]

Corresponding revision can be found on line 108 and 125~130 of page 5.

2.  For "the annual precipitation in China since 1980" in Section 2.1, are there several annual precipitation values? Are the six indicators at the national-scale or the city-scale?

Response: Thanks for the comment. "The annual precipitation since 1980" is replaced by the annual precipitation calculated by aggregating GPM data in 2017. The six indicators are at the city level. Corresponding revision can be found on line 90~95 of page 3.

3.  The definition of the stable TLR number should clarify in 2.2.1.

Response: Thanks for the comment. We add the definition to the sentence in 2.2.1. *"The stable grids are the regions with stable human activity and rhythm in urban area."* Corresponding revision can be found on line 109 of page 5.

4.  In Section 2.2.4, it's not clear for me for the definition of rainfall threshold.

Response: Thanks for the comment. The rainfall threshold is the peak intensity of the rainfall event which triggers collective human activity anomalies. Corresponding revision can be found on line 62~72 of page 2.

5. Fig.7 shows the regression coefficients for the six indicators, however, the description of the figure focuses on the correlation coefficients. It confuses me as they are different.

Response: Thanks for the comment. We modify the correlation coefficients to regression coefficients. Corresponding revision can be found on line 287 and 297 of page 13.

6. I am struglling to follow the manuscript, but I cannot know the scale of the analysis in 2.1. If it is the city level, how to transform the TLR and GPM grids into it?

Response: Thanks for the comment. We reorganized the structure and content of the methods. The section 2.2.1 represents the transform of TLR data from the grid scale to the city scale. We firstly employed the S-H-ESD method to detect anomalies from the gridded TLR time series. Then we extracted the total numbers of the grids with positive (PTLR) and negative anomalies (NTLR) by city, respectively and then examined the variations in the PTLR/NTLR time series over the periods with rains and without rains to identify whether a rainfall event triggers collective human activity anomalies.

The section 2.2.2 represents the transform of GPM data from the grid scale to the city scale. We first extracted the hourly rainfall intensity for each city/hour, i.e., the average hourly GPM precipitation within the stable grids of the city. And then, the rainfall events for each city are extracted from the hourly rainfall intensity time series. Corresponding revision can be found on page 6 and 7. Corresponding revision can be found on line 110~140 of page 5.

7. There are several citation errors. For example, "activities(Jiawei Yi et al., 2019;" should be "activities(Yi et al., 2019;","the method Qian et al.(Jiale et al., 2021) proposed." should be "the method proposed by Qian et al. (2021)."

Response: Thanks for your valuable suggestion. All issues about citation in the text have been corrected. Corresponding revision can be found on line 44 and 109 of page 2 and 5.

This study quantified the responses of collective human activities to rainfall events using used the Tencent location request data, and evaluated the variations in the responses in China with the aim to explore the inequality of urban resilience in China. The topic is very interesting and this study has remarkable significance for emergency response and disaster management. While some revisions are still needed to improve the manuscript:

1. The title is not accurate; it should be the resilience (or response) to rainfall not to rain.

Response: Thanks for your valuable suggestion. We have modified the title to *Quantifying unequal urban resilience to rainfall across China from location-aware big data*.

2. The Introduction section could be better organized to clarify the scientific question (or research gap) and aims of this study.

Response: Thanks for your valuable suggestion. We reorganized the research gap and the aims of this study to the Penultimate paragraph in section 1. Specific information is as follows:"*However, previous resilience metric, which mainly forces on unique disaster event, is not suitable for the assessing in large scale. Two resilience metrics were introduced into this study from other fields. The sensitivity is a widely used tool for understanding resilience in different regions in many other weather events, such as heat wave and air pollution (Hong et al., 2021b; Wang et al., 2021). For example, Zheng et al. (2019) defined the links between the city-level happiness index calculated from the social media data and the daily local air quality metric as the perception sensitivity and explored its spatial variation. However, response sensitivity is not yet to be studied for rainstorm events through analysing the relation between the city-scale human activity response metric and rainstorm event index. Another index, rainfall threshold, is commonly used to study rainfall events that have resulted in landslides (Marra et al., 2016; Naidu et al., 2018). In this study, rainfall threshold, which is defined as the minimum rainfall index that corresponds to significant urban human activity response anomaly, is introduced to the study of the urban resilience. Two metrics can effectively depict the urban resilience in different focuses.*" Corresponding revision can be found on line 62~72 of page 3.

3. Urban resilience is very broad concept with many different elements and properties, mainly related to the capacity, sensitivity, flexibility of urban systems (including the community, infrastructure, institution, etc.). So, what does urban resilience mean in this study? How can it be related to human activities? Moreover, the rationality for using rainfall threshold and response sensitivity to describe urban resilience needs more justification.

Response: Thanks for your valuable suggestion. We give a full answer to this question in question 3. Corresponding revision can be found on line 62~72 of page 3.

4. I would suggest to add a discussion for the limitations of this study and the prospect for future study at the end of the manuscript.

Response: Thanks for your valuable suggestion. We add a discussion for the limitations of this study and the prospect for future study at the end of the manuscript. Specific information is as follows: "*The study could by further studied. Rather than all the residents of a city, the Tencent location request dataset is generated by over one billion monthly active users. The Tencent dataset's aggregate geotagged human activities may underestimate the effects of rainstorms on infrequent users, particularly the elderly and children. To address this limitation and further investigate human responses to various*

*weather events, our future studies would aim to integrate multisource geospatial datasets. Furthermore, identifying disaster types such as rainstorm, waterlogging, and flood from social media data and then analyzing regional response variation of large-scale human activity in different disasters can improve deep understanding of urban resilience"*. Corresponding revision can be found on line 340~347 of page 15.

5. It is not clear how the cities were classified to different types (e.g., HL, ML, HM, LL)?

Response: Thanks for your valuable suggestion. We add the classification method to the section 2.2.4. "*Finally, we separately classified the rainfall threshold and response sensitivity indices of the 346 cities into three classes using the Jenks natural breaks classification method, which clusters data into different groups by seeking minimum variance within a class and maximum variance between classes(McDougall and Temple-Watts, 2012).* " Corresponding revision can be found on line 172~175 of page 7.

6. There are several writing errors, such as "the method Qian et al.(Jiale et al., 2021) proposed"(Page 4), "Zou et al.(Zou et al., 2018) used" (Page 14).

Response: Thanks for your valuable suggestion. All issues about citation in the text have been corrected. Corresponding revision can be found on line 44 and 109 of page 2 and 5.

7. The supplementary Fig. 6 should be put in the manuscript rather than in the supplementary, as it appears for many times and is vital for the understanding of how rainfall time, peak intensity and duration affect human activities.

Response: Thanks for your valuable suggestion. We have put the supplementary Fig. 6 to the right position. Corresponding revision can be found on line 218 of page 9.

This paper uses the location-aware big data from the Tencent Chinese social media platform to explore the spatial distribution of urban resilience in China. The paper is interesting, the link rainfall intensity and urban resilience it is a very topical problem. The scope of the study is significance for emergency response, and it investigate a very large area. The text is well-organized and well-written.

I have some general comments:

1. Urban resilience is a very complex concept. I can't find how the author define urban resilience in this study and how can they relate resilience to the anomalies in human activities induced by the heavy rainfall. Please, try to explain better.

Response: Thanks for your valuable suggestion. We reorganized the definition of urban resilience and the relationship between the resilience and the human activities to the Penultimate paragraph in section 1. Specific information is as follows:"*Urban resilience refers to the ability of an urban system to prepare for, respond to, and recover from adverse events(Ambelu et al., 2017; Hong et al., 2021a; Liao, 2012; Meerow et al., 2016). For example, Hong et al., (2021b) quantified changes of mobility behaviour before, during, and after the Hurricane Harvey using smartphone geolocation data, and analysed the spatial variable of community resilience capacity which was defined as the function of the magnitude of impact and time-to-recovery. Human activities may also change in response to mild yet frequent adverse natural events, such as urban rainstorms. Unlike Hurricanes, dwellers are usually not mobilized by relevant authorities to prepare for and resettle after such events. Instead, nearly 90% of flood-related tweets in a city are released during heavy rains (Wang et al., 2020). Consequently, human activities mainly show how an urban system respond to but not prepare for and recover from such adverse natural events (Qian et al., 2022; Zhang et al., 2022). As a result, urban resilience to mild and frequent adverse events refers to the ability of an urban system to respond to adverse events.*" Corresponding revision can be found on line 31~32 and 50~60 of page 2.

2.  The authors cite the supplementary material as fundamental part of the manuscript. Please select the figures you consider to be important and try to add to the text (as for example figs 4 and 6).

Response: Thanks for your valuable suggestion. We have put the supplementary Fig. 6 to the right position. Supplementary Fig. 4 is not added to the text because Fig. 3 (a) presents the result for the same topic. Supplementary Fig. 4 is the complement of Fig. 3(a). Corresponding revision can be found on line 218 of page 9.

3.  The cities classification into different types HL, ML, HM, LL is not described, while it is very important for the discussion section

Response: Thanks for your valuable suggestion. We add the classification method to the section 2.2.4. *"Finally, we separately classified the rainfall threshold and response sensitivity indices of the 346 cities into three classes using the Jenks natural breaks classification method, which clusters data into different groups by seeking minimum variance within a class and maximum variance between classes (McDougall and Temple-Watts, 2012)."* Corresponding revision can be found on line 172~175 of page 7.

4.  I would suggest to add a discussion regarding limitations and future perspectives of this study since the authors do not investigate some important relation between the physical factors and the human activities. For example, it could be crucial to relate the indices the author found with physical data, altitude of the city, the average slope, while for the human activities they could investigate the number of emergency call, or the number of car accident, for citing someone. Please add a legend with acronyms explanation

Response: Thanks for your valuable suggestion. We add a discussion for the limitations of this study and the prospect for future study at the end of the manuscript. Specific

information is as follows: *"The study could by further studied. Rather than all the residents of a city, the Tencent location request dataset is generated by over one billion monthly active users. The Tencent dataset's aggregate geotagged human activities may underestimate the effects of rainstorms on infrequent users, particularly the elderly and children. To address this limitation and further investigate human responses to various weather events, our future studies would aim to integrate multisource geospatial datasets. Furthermore, identifying disaster types such as rainstorm, waterlogging, and flood from social media data and then analyzing regional response variation of large-scale human activity in different disasters can improve deep understanding of urban resilience"*. Corresponding revision can be found on line 340~347 of page 15.